# Ontology-Based Regression Testing: A Systematic Literature Review

**Muhammad Hasnain [1,\*], Imran Ghani [2], Muhammad Fermi Pasha [1] and Seung-Ryul Jeong [3]**

1 School of Information Technology, Monash University, Subang Jaya 47500, Malaysia; muhammad.fermipasha@monash.edu
2 Computer and Information Sciences, Virginia Military Institute, Lexington, VA 24450, USA; ghanii@vmi.edu
3 Graduate School of Business IT, Kookmin University, Seoul 136, Korea; srjeong@kookmin.ac.kr
\* Correspondence: Muhammad.malik1@monash.edu

**Abstract:** Web systems evolve by adding new functionalities or modifying them to meet users' requirements. Web systems require retesting to ensure that existing functionalities are according to users' expectations. Retesting a web system is challenging due to high cost and time consumption. Existing 'systematic literature review' (SLR) studies do not comprehensively present the ontology-based regression testing approaches. Therefore, this study focuses on ontology-based regression testing approaches because ontologies have been a growing research solution in regression testing. Following this, a systematic search of studies was performed using the "Preferred Reporting Items for Systematic Reviews and Meta-Analyses" (PRISMA) guidelines. A total of 24 peer-reviewed studies covering ontologies (semantic and inference rules) and regression testing, published between 2007 and 2019, were selected. The results showed that mainly ontology-based regression testing approaches were published in 2011–2012 and 2019 because ontology got momentum in research in other fields of study during these years. Furthermore, seven challenges to ontology-driven regression testing approaches are reported in the selected studies. Cost and validation are the main challenges examined in the research studies. The scalability of regression testing approaches has been identified as a common problem for ontology-based and other benchmark regression testing approaches. This SLR presents that the safety of critical systems is a possible future research direction to prevent human life risks.

**Keywords:** software testing; semantic rules; scalability; surveillance systems; security; validation of test cases

## 1. Introduction

Regression testing is a relevant research field focused on ensuring software works correctly after being modified or when new functionalities are added. The widespread use of information technology techniques has enabled the rapid development of applications. However, modifications or the addition of new features in the applications require retesting the web applications, which adds the problem or concern for software testers to retest the applications. Retesting web services/web applications is a relatively hard job for testers, and it also incurs a high cost. 'Regression Testing' (RT) with its subtypes 'test case selection', 'test case reduction', and 'test case prioritization' have been widely examined in research studies [1]. Test case selection approaches aim at increasing the effectiveness based on their measurement capabilities, including the cost, coverage, and fault detection [2–4]. Test case prioritization (TCP) approaches let researchers' reorder test cases to reveal maximum faults based on specific criteria. Test cases with the top defects are tested before the test cases with fewer failures [5]. Proposed RT approaches rely on some criteria, and the ontology of systems is one of these criteria proposed to perform test case selection, reduction, and prioritization.

Ontology refers to basic concepts and their relationship in a research domain [6]. Due to ontology reasoning, the ideas which are harder for their explicit expression are derived. Ontology provides further analysis and assessment knowledge, including the accurate classification of faulty and non-faulty software engineering modules. The field of ontology covers the aspects of methodologies and metrics applied to solve issues in software engineering [7]. OntoClean is one of these methods that covers the use of some properties, including unity, rigidity, identity, and dependency [8]. The OntoClean method, along with meta-properties, is proposed to provide entities' semantic and logical meaning. In [9], researchers made extensions to define ontology by expressing vocabulary concepts and their use. In this entire document, web applications and web services are interchangeably used.

The motivation behind this SLR is to investigate the continuous lack of discussion on ontology-derived regression testing in several SLRs [10]. Ontology-derived testing can be considered under the domain of regression testing [11]. Therefore, first, we present an overview of existing SLRs on regression testing and ontologies.

Table 1 is the illustration of SLRs in the area of regression testing. The first SLR by Qui et al. [12] includes two papers discussing web services' semantic behavior. Studies [13,14] did not include papers on ontology-based regression testing. A study by de Souza et al. [15] presented a description of web services ontologies. They include two ontology-driven models, 'web services modeling ontology' (WSMO) and 'web services description language semantics' (WSDL-S). Therefore, our SLR's motivation is to help researchers and software practitioners working on the ontologies in regression testing by presenting the state-of-the-art ontology-based RT approaches. We are also introducing the first SLR on ontology-based regression testing approaches.

**Table 1.** Existing SLRs on regression testing.

| Sr. No | Research Focus | Ontology-Based Regression Testing | Number of Scholarly Studies | Reference |
|:---:|---|:---:|:---:|:---:|
| 1. | Regression Testing | x | 159 | [9] |
| 2. | Regression test case prioritization | x | 65 | [10] |
| 3. | Test case prioritization | x | 120 | [11] |
| 4. | Regression testing of web services | √ | 30 | [12] |
| 5. | Regression test case selection | x | 47 | [3] |
| 6. | Test case prioritization | x | 69 | [13] |
| 7. | Test Case prioritization of systems | x | 90 | [14] |
| 8. | Testing of semantic web services | √ | 43 | [15] |
| 9. | Agent-based test generation | x | 115 | [16] |

The structure of this paper is as follows.

First, the authors present a literature review in Section 2. Further, they present the research methods to conduct the SLR on RT and ontologies in Section 3. Next, they show results and their discussions in Section 4. Finally, this SLR study's research limitations are presented in Section 5, while researchers conclude the SLR and present future implications in Section 6.

## 2. Literature Review

This section presents an overview of the existing literature on software testing regarding ontologies.

In a recently published study, Akbari et al. [17] focused on the integration testing product lines. They claimed that the cost of integration testing was a real challenge that required the attention of researchers. They proposed a 'feature model' (FM) to support the

PINE method to examine integration testing costs. They reused the domain engineering artifacts to prioritize and execute the integration test cases. Their research work helped reduce 82% cost on integration testing and detecting 44% of integration faults. However, regression testing has been further explored in the following studies.

Hemmati [18] stated that many proposed web services TCP approaches showed limitations due to the high coverage of large-scale web services under regression testing. Hillah et al. [19] discussed service integration and emphasized corrective and perfect web services maintenance. The former type of maintenance is used for bug fixing, and the latter one typically involves the addition of new operations or already planned extensions. To meet users' requirements, web services evolve and need retesting based on the scalability behavior of web services. Therefore, regression testing involves the scalability factor in designing the approaches.

The scalability issue of test case prioritization strategies was highly emphasized in a study [20]. Researchers in the same research suggested future research works on extensive web services regarding cost-saving. Another study [21] presented their proposed 'dynamic symbolic execution' (DSE) approach efficiently by analyzing the code. However, it has limited scalability due to path explosion issues. In the same study, researchers explicitly considered mitigating the limitation of the DSE approach. To further explore the use of ontology in testing, Zhu and Zhang [22] proposed a 'Software Testing Ontology for WS' (STOWS) approach to manage the semantics of web services, their registry and discovery. The authors pointed out the problem of integrating services from different owners. A collaborative testing approach was proposed to overcome this issue. The proposed approach used ontologies to support the wide range of testing activities. The same research proposed to use the crowdsourcing approach to add new vocabulary and update it for public information. The STOWS approach is scalable as a test broker can handle significant test problems for registered testers. However, the proposed approach did not reveal the risk factors in designing the approach. To overcome this issue, Wang et al. [23] proposed a risk-based TCP by keeping in mind the existing TCP approaches, which use the execution information of test cases and the history of code changes. However, researchers in the above-mentioned study examined assigning the weights to classes regarding the complexity of semantic web services. By considering the system topology, we can design the information flow in the ontology of the system. To document the classes and their relationships, involving coverage and fault information may reduce the efforts to order the test case of large-size systems.

De Souza et al. [24] verified that developing ontology for multiple web services was challenging. Before this study, researchers in [25] pointed out that capturing changes in requirements of web services was an exact moment to use ontology because information about the last modifications is further used in regression testing.

Bai and Kenette [26] emphasized group testing to improve test efficiencies and reduce testing costs for comprehensive testing of web services. They proposed a risk-based approach to categorize and reorder test cases using web services' target features to detect faults earlier in web services. Therefore, as mentioned above, the researchers' effort, as mentioned in the study, is considered among the initial research studies that have undertaken TCP regarding ontology to quantify the risk assessment. Failing web services is mainly due to data misuse, failed service binding, and unanticipated usage scenarios.

## 3. Research Method

We used a research method following the PRISMA guidelines [27]. This method has been reported with a four-phase approach in the literature. These four phases include identification, screening, eligibility, and included (See Figure 1).

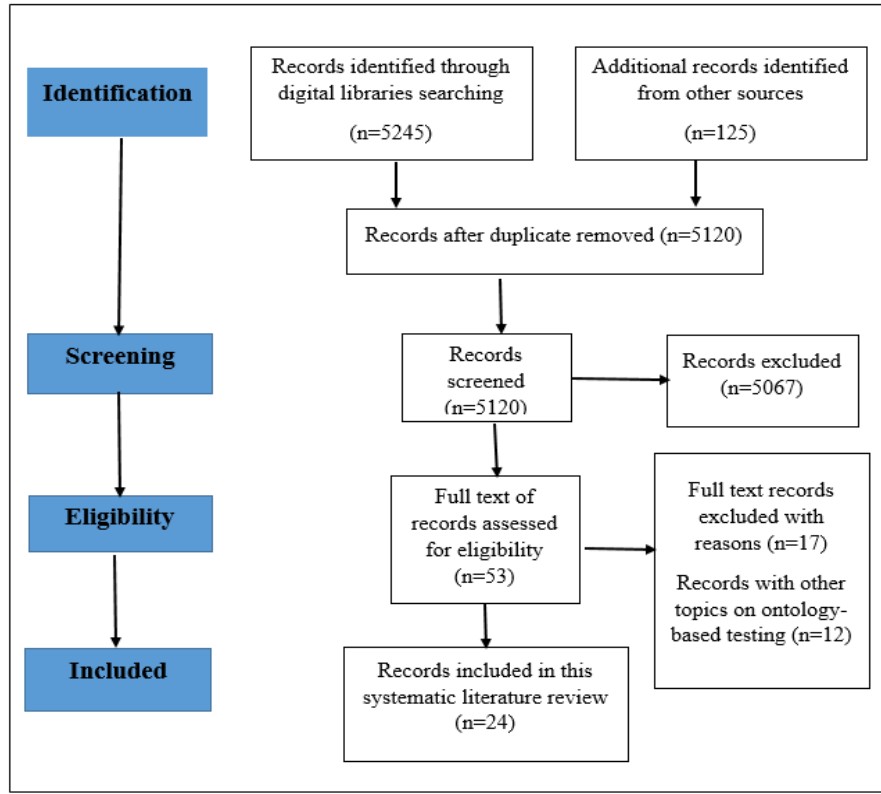

**Figure 1.** Study selection process [27].

*3.1. Research Questions*

Below are the research questions studied for this SLR (see Table 2).

**Table 2.** Proposed research questions.

| RQ ID. | Description |
|---|---|
| RQ1 | What is the roadmap of regression testing? |
| RQ2 | What are the state-of-the-art techniques of ontology-based regression testing? |
| RQ3 | What are the challenges and limitations of current approaches? |
| RQ4 | What are the possible future research directions? |
| RQ5 | What are the unique issues of ontology-based regression testing compared to other regression testing approaches? |

*3.2. Search Keywords*

A set of keywords was constructed using the words and phrases from research questions. Search strings relevant to the topic of interest and research questions were formulated [28]. We used well-known research repositories, including IEEE Xplore, ACM digital library, ScienceDirect, Springer Link, Web of Science, and Wiley & Sons, to search the research articles published between 2007 and 2019. The search strings applied to the digital repositories mentioned above are shown in the following Table 3.

**Table 3.** Search strings used for paper's retrieval.

| Data Repository Name | Search String |
|---|---|
| IEEE Xplore | (((((("All Metadata": Ontology) AND "All Metadata": Regression Testing) OR "All Metadata": Test Case Prioritization) AND "All Metadata": Test Case Selection) AND "All Metadata": Test Case Generation) |
| ACM digital Library | (+Ontology + Regression + Testing + Test + Case + Prioritization + Test + Case + Selection + Test + Case + Generation) |
| ScienceDirect | "Ontology, regression testing, test case prioritization, Test Case Selection, Test Case Generation." |
| SpringerLink | 'Ontology AND "Regression Testing" AND (Test OR Case OR Prioritization, OR Test OR Case OR Selection, OR Test OR Case OR Generation)' |
| Web of Science | (Ontology) AND TOPIC: (Regression Testing) OR TOPIC: (test case prioritization) OR TOPIC: (Test Case Selection) OR TOPIC: (Test Case Generation) |
| Wiley & Sons | "Ontology" anywhere and "Regression Testing" anywhere and "Test Case Prioritization" anywhere and "Test Case Selection" anywhere and "Test Case Generation". |

It was found that when we applied "regression testing" OR "test case prioritization" strings, many irrelevant studies to the topic of this SLR were returned. We refined our search strings using Boolean 'AND' and 'OR' operators between search keywords to find relevant publications.

### 3.3. Selection of Research Publications

The obtained research documents were examined by two independent reviewers (MH and IG). This phase mainly aimed at verifying the research articles, which were potentially eligible and could be used for further analysis. At the screening phase, research documents were screened via title, abstract, and keywords. However, two reviewers (MFP and SRJ) read and reviewed full-length documents resulting from the screening phase at the eligibility phase. Since ontology has been used in software testing, reviewers included papers related to the topic "ontology-based regression testing" with great caution in this systematic review. Therefore, 24 research documents were included in this study.

### 3.4. Inclusion and Exclusion Criteria

The electronic database search process resulted in the selection of 53 studies. The selection of the studies underwent the review of titles, abstracts, and keywords. Next, we performed a full-length review of studies. As a result, 24 studies were finally selected for this SLR. Studies that had matched search strings with the titles, abstracts, and keywords were included in the first step of the selection process. The authors leveraged the guidelines and recommendations made by researchers [27,29].

We included peer-reviewed articles on the topic "ontology-based regression testing" and further explore the studies' inclusion and exclusion criteria as follows.

The inclusion criteria of research studies are defined based on the titles, abstracts, and keywords as follows:

➢ Publications that discuss ontology-based testing of web services were included.
➢ Publications that discuss ontology-based regression testing of web services were included.
➢ Publications that discuss the ontology-based test case prioritization techniques were included.
➢ Publications that discuss issues/challenges regarding ontology-based regression testing were included.

Next, the authors defined the exclusion criteria of research articles as given in the following:

➢　　Duplicate papers on the research topic were excluded.
➢　　Publications that did not discuss ontology-based regression testing of web services were excluded.
➢　　Publications that did not provide a technical discussion on the research topic of this SLR were included.
➢　　Publications written in a language other than English were excluded.

### 3.5. Quality Assessment

In the final step of the research method, we evaluated the chosen 24 studies based on the Quality Assessment Criteria (QAC). The main objective of the QAC was to decide on the quality of identified research articles to ensure the worth of their findings and interpretations. We proposed five QAC questions, which are given in the following Table 4.

**Table 4.** Quality assessment criteria and proposed questions.

| QAC Question No. | Description |
| --- | --- |
| Q1 | Does the research topic is relevant to ontology-based regression testing? |
| Q2 | Does the research study has a clear context regarding the research topic? |
| Q3 | Does the research sufficiently define research methodology? |
| Q4 | Is the data collection process effectively revealed? |
| Q5 | Is the data analysis method appropriately explained? |

The authors adopted three quality rankings, 'high', 'medium', and 'low' from the study [30]. The rankings were based on the scoring procedure used to evaluate every QAC as given:

$$Yes(Y) = 1, Partly(P) = 0.5 \text{ and } No(N) = 0.$$

## 4. Results and Discussion

This section of the SLR presents the chosen studies' outcomes, which were found relevant to answer the proposed RQs in this SLR.

### 4.1. RQ1: What Is the Roadmap of Regression Testing?

The below figure provides a roadmap of the regression testing approaches tailored towards the fault detection's targeted objectives in software systems.

Figure 2 presents the roadmap of the regression testing to identify the faults in the modified systems. The roadmap has six entities, along with several activities. As shown in Figure 2, the first layer indicates benchmark techniques in the research area of regression testing. We can see code coverage-based [31], requirement-based [32], similarity-based [33], session-based [34], ontology-based [35], and location-based [36] RT approaches in the roadmap as shown in Figure 2. In the second layer, the information used for the proposal of each benchmark technique is presented. For instance, code coverage and requirement-based RT techniques are proposed by code and requirements information, respectively, and so on. In the third layer, the benchmark techniques' main criteria show that faulty code and requirement priorities are widely employed in code coverage and requirement-based RT techniques. At the fourth level, the next level of proposed criteria is defined regarding RT approaches. At the fifth layer of the roadmap, we have presented the benchmark RT techniques' ultimate objectives. Overall, the aim of RT techniques remained test case generation, test case execution, and fault identification in the web systems.

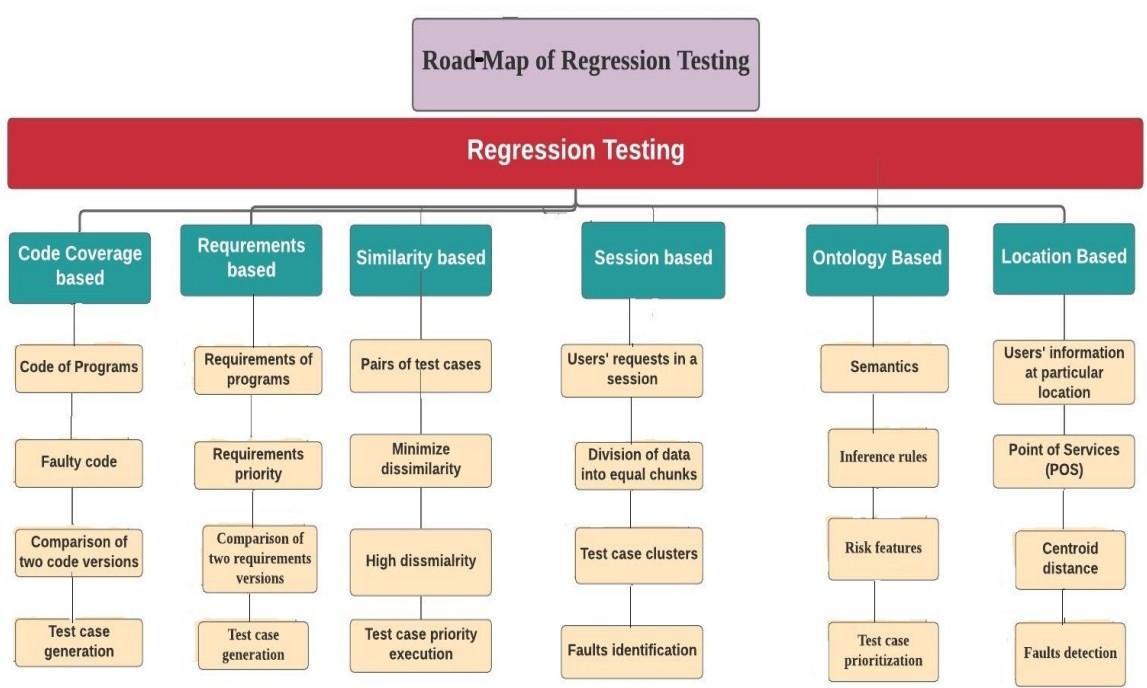

**Figure 2.** Roadmap of regression testing approaches.

*4.2. RQ2: What Are the State-of-the-Art Techniques of Ontology-Based Regression Testing?*

This section presents an overview of state-of-the-art ontology-based regression testing. The study's inclusion criteria show the chosen studies' year-wise distribution in Figure 3.

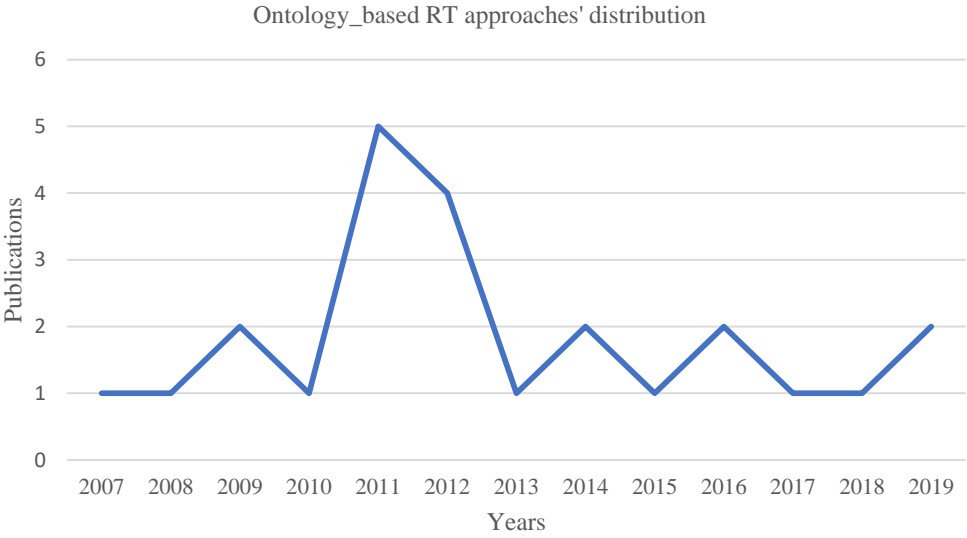

**Figure 3.** Year-wise publications on ontology-based regression testing.

Figure 3 shows that most of the research studies were published in 2011 and 2012, respectively. We can see that the number of research studies on TCP was in limited numbers other than in the years 2011–2012. The research community remained more focused on ontology-based regression testing for test case generation. In the following, we discuss the research studies on the ontologies in regression testing.

Researchers in [37] proposed the severity-based TCP technique, which detected more faults than other TCP techniques. However, this study does not discuss the criteria that have been applied to determine fault severity. Therefore, we need to incorporate risk

metrics to make ontology-based TCP more explicit in achieving results. Consequently, we may add new components to the existing testing model [38], which compares existing TCP approaches and their efficiencies.

The research studies have widely discussed the probability of running services and related risks [39,40]. As earlier mentioned in studies, ontologies have been used to represent dependency relationships, i.e., classes and links. Mathematical models have been significantly applied for measuring the failure probabilities of web services. Static web applications that have been updated or modified have been missed regarding the above studies' research efforts. Although the adaptive approach of risk estimation and test case ranking is of great importance, it would be better if researchers discussed static web services regarding ontology-based TCP.

The study [41] focused on ontology-driven systems for regression testing. The study involved targeted ontologies rather than using the code of programs. The researchers compared the ontology of the old system to the ontology of the updated or modified system. Thus, the newly added or changed elements of an ontology helped in identifying the potential test cases. In this way, they evaluated the proposed ontology-based regression testing technique by implementing it in the ONTORETEST tool. A graph was built based on the ontologies of old and modified systems [42]. As a result, cost and time factors were reduced from the ontology-based RT approach because this research had executed test cases of only changed parts to identify faults.

Inference rules and ontologies present a better solution to testing issues in the area of regression testing. This claim was proved by implementing the inference rules on the Prologue tool's ontologies [35]. Thus, OWL functional tests are translated to the prologue syntax. Researchers compared the manually created test cases to their corresponding requirements and observed inconsistencies. Then, they formulated a set of inference rules in plain English. Ontologies are also used to perform testing in safety-related web systems.

The safety analysis report (SAR) document was a credible source for the automated validation of TCG for the nuclear industry systems [43]. Accidents and associated safety measures are crucial elements enclosed in the proposed approach. SAR ontology is applied to the XML markup language and safety analysis markup language (SAML). The latter study's main contribution is developing the TCG tool, which automatically performs addition, deletion, and saving the tags in the markup language. Most critical systems, including the nuclear safety system, need to comply with users' needs. To validate the nuclear system, the study [44] proposed ontology-specific knowledge to validate it systematically. The proposed methodology in the studies, as mentioned earlier, is similar because both studies use SAR to convert scenarios in the processing structures.

In the following Table 5, we summarize the studies that focused on ontologies in proposing approaches in regression testing.

Table 5 shows us ontology-based studies and authors' information, problems, proposed techniques, advantages, and limitations of the proposed approaches. The proposed approaches mainly addressed cost, testing in an open platform, modification in evolving web systems, and validation of test cases in nuclear safety systems. This SLR shows the remarkable penetration of ontologies in the prioritization of test cases in web services. Ontology-based regression testing approaches were found effective in saving costs and time compared with random testing approaches. Semantics-based designing of approaches can be further used with the help of 'artificial intelligence' to monitor the continuous performance of web services.

**Table 5.** Summary of ontology-based approaches.

| Authors | Problem | Proposed Technique | Advantage | Limitation |
|---|---|---|---|---|
| Askarunisa et al. [37] | High cost on web services testing | An automated testing framework | Effective prioritization of web services | No cost resources were used in the proposed approach |
| Askarunisa et al. [38] | An increase in cost due to source code unavailability | Semantic-based protégé tool | Faults detection is better than the traditional approaches | Coverage criteria is not clearly mentioned |
| Bai et al. [40] | Web services testing in an open platform | A risk-based approach using semantics | Faults detection with a high impact | The reliability of web services is not mentioned |
| Kim et al. [41] | Handling changes in the evolving system is a difficult task | Regression test selection | Reduces the overall time to rerun test cases | This approach might have missed some necessary test cases |
| Fan et al. [43] | Validation of tests in an ad hoc fashion | Domain-specific ontology-based approach | Systematic generation of TCG from safety analysis report | The cost to ensure the safety of a web system is not mentioned |
| Tseng et al. [44] | Uncertain test coverage from random testing | Domain-specific ontology using requirements | Ensures users' safety, more effective than random testing | Validation of web systems at large-scale systems is not done |

### 4.3. RQ3: What Are the Challenges and Limitations of Current Approaches?

To answer RQ3, the authors presented the challenges and limitations identified in the ontology-based regression testing approaches. These limitations and challenges are given as follows:

- Validation of approaches. Although test case grouping presented in [39] helps remove the failed services, it still shows limitations in prioritizing test cases as low prioritized test cases might be grouped with the high prioritized test case [43]. Grouping high and low priority test cases is justified and plausible only if it is supported by previously evaluated criteria [22,42].
- Cost. To separate the failed test cases, all test cases of an application are rerun, which is expensive regarding the testing cost [45,46]. Adaptive testing is a steady job that requires sources for testing web services. The cost issue is widely identified in ontology-based studies [41,42,47].
- Scalability. The scalability and applicability of the proposed TCP approaches on real-world web applications are other limitations of the existing research studies [22,39,40] regarding ontology-based TCP in regression testing.
- Risk assessment. Application output behavior and risky scenarios are also not addressed in these research studies [22,39]. Study [39] presented a model based on the semantics from the workflow of web services, ontology usage, and ontology dependency to identify the contributing risks during the execution of web services. At the same time, the risk of errors was not taken into account while web systems were updated during the execution of tasks. To prevent or reduce such errors is an open question for researchers [22].
- Reliability. Web systems' reliability in terms of regression testing is missed in the existing studies.

A study [40] proposed a risk assessment approach for the ranking of test cases. Researchers in this study considered risk assessment in the context of users' input/output and pre/post conditions, processes, and ontology constraints. The risk assessment covers the dissatisfaction of regulatory/legal requirements and relates mainly to a process, work, or project that does not achieve the expected functional behavior. The proposed TCP approach does not involve the reliability of web services. A product risk carefully refers to

security and safety behaviors [48], as well as it also concerns the reliability behavior of the web services. Therefore, web services' reliability factor can improve regression testing and prioritize test cases. In addition to the challenges mentioned earlier, the failure of critical systems is closely related to the proposed maintenance approach [49], which cannot detect earlier faults. It is still challenging for researchers to optimize the test case selection and TCP in regression testing.

### 4.4. RQ4: What Are the Possible Future Research Directions?

Based on this SLR, we present a summary of future research directions.

The ontology-based test case reduction approach is efficient for the regression testing of systems in the bioinformatics domain [41]. Thus, we can evaluate the proposed approach on systems from other research areas. The proposed method can be extended to perform TCP, test case augmentation and identification, and test repairing, which are vital for the successful running of a system [50,51]. The second research direction emerges from studies [52,53], where researchers propose rule-based regression testing approaches and tools. Future research can present a coherent framework by integrating TCG, mutation testing, and feedback on testing results. It may enable the output checking correctness regarding users' requirements.

Moreover, we noticed that the simulation environment is adequate to verify the proposed approaches [43]. Since the proposed techniques deal with the automated validation of TCG, a dedicated simulator is more specific for safety-critical systems. In addition, human lives are the main entities to be saved during daily operations in workplaces. Particularly, workplaces that integrate with the automated critical systems require to be made safe for their surroundings. Moreover, dynamic testing of autonomous deriving functions at the industrial level is another future direction highlighted in a study [49]. It is suggested that static parts such as roads and weather can be integrated with ontologies to extract better constraints, i.e., surveillance systems and their security.

### 4.5. RQ5: What Are the Unique Issues of Ontology-Based Regression Testing Compared to Other Regression Testing Approaches?

Before our SLR, de Souza et al. [15] identified the research gap that the model-based testing has not dealt with because abstract test cases are trivial to map them to executable and concrete test cases. Compared with the paper [22], scalability is a focused research problem regarding test brokers because they face re-execution of test cases of a suitable size. The scalability issue remains a commonly identified challenge for both ontology-based RT and collaborative testing of web services. Due to web services' evolution, scalability is still an open issue in the software testing domain. Validation of proposed RT approaches is mentioned as a problem [50] because all proposed simulation-based approaches do not address the testing of web applications in real-world scenarios. Therefore, the validation problem is close to our identified problem. A study [19] stated that cloud testing faces the scalability challenge because distributed systems need enhancement in their performance over time. This performance enhancement can combat the increasing workload on web services. Finally, inadequate validation of test cases is a unique challenge for automated test case generation [51]. Validation of the test cases is crucial for real-time processing to reduce software testing costs and effort.

## 5. Limitations of the SLR

There are several limitations of this SLR, which are too vital to note. In turn, these limitations can provide research directions to researchers and practitioners. Since the literature search was focused on ontology and RT and its techniques, future works may broaden the research strategies to identify new insights regarding ontology and its use in the software engineering domain. We reviewed RT techniques concerning the ontologies; we could only find limited research articles, and therefore, this research area may be further explored in future works.

## 6. Conclusions and Future Implications

This SLR identified ontology-based RT approaches proposed between 2007 and 2019. Most of the studies focused on TCG and TCP research topics. Our SLR presented the roadmap by identifying the benchmark approaches along with their objectives. This study identified six challenges of ontology-based RT approaches. The cost and validation were difficulties widely examined in the selected studies. Scalability, risk assessment, failure of critical systems, and reliability are still open regression testing issues. Our SLR presented the effectiveness of ontology-based RT approaches for addressing the failure in systems other than bioinformatics. This SLR's findings suggested developing more specific simulation tools regarding the security of the critical system in future works.

**Author Contributions:** Conceptualization and methodology, M.H.; software, I.G.; validation, and formal analysis, M.F.P.; investigation, M.H.; resources, M.F.P.; data curation, I.G.; writing—original draft preparation, M.H.; writing—review and editing, M.F.P.; visualization, S.-R.J.; supervision and project administration, S.-R.J.; funding acquisition, I.G. All authors have read and agreed to the published version of the manuscript.

**Funding:** This work was supported in part by the Commonwealth Cyber Initiative, an investment in the advancement of cyber R&D, innovation, and workforce development. For more information about CCI, visit www.cyberinitiative.org (accessed on 14 October 2021).

**Acknowledgments:** As a corresponding author, I wish to express my sincere gratitude to all of my colleague authors who helped me write this SLR paper.

**Conflicts of Interest:** The authors declare that they have no conflict of interest.

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
