# Peer review of "Ontology-Based Regression Testing: A Systematic Literature Review"

_applsci, doi:10.3390/app11209709_

Round 1

Reviewer 1 Report

This paper identifies ontology-based regression testing approaches proposed between 2007 and 2019. Most analyzed of the studies focused on  TCG and TCP  research topics.  The review presents the roadmap by identifying the benchmark approaches along with their objectives. The authors study the challenges of ontology-based regression testing approaches. Moreover, the cost and validation are examined in the selected studies. 

The parts marked in green complement the paper well. 

The paper stands out for a reason. The authors clearly explain the challenges to ontology-driven regression testing approaches.
Second, the authors examine cost and validation in the research studies.

Therefore, I consider this paper ready for publication.

Author Response

Thank you for your time and feedback about reviewing our paper. 

Reviewer 2 Report

The authors review ontology-based regression testing studies using the PRISMA methodology seems sound. However, there are major issues that the authors need to adress to improve the quality of the article.

* Line 37: Abbreviations in abstract are in the form <long-name> (<short-name>). In this line is in the form '<long-name>' short-name. The authors are suggested to change it.
* Line 101 and 102: "To develop a complete testing ontology, we require a new testing approaches and methods". This seems so disconnected from the paragraph about Zhu and Zhang's work. Is this sentence something you (the authors of this SLR) state or is it something Zhu and Zhang state? I think the sentence and maybe its position in the paragraph must be reconsidered.
* Line 109: "system a topology"?
Sentence from lines 158 to 160 is hard to understand.
Figure 3 has a bad quality and could be improved. Some lines are duplicaded and some line angles (corners) are unecessary.
Roadmap or road map?
* "We can see that the number of research studies has been limited to TCP in regression testing." However, Figure 4 only shows an aggregation of ontology-based RT approaches. Therefore, I am not sure that the reader can "see" that studies have  been limited to TCP.
* It seems that some paragraphs are so disconnected from the previous ones and that the flow of the narrative is a little bit chaotic.
* This reviewer thinks that the authors have done a great work in reviewing the literature and applying the PRISMA methodology. However, the contribution of their work is hidden due to poor connection between the units of thought (sentences not properly fit to paragraphs and connection between paragraphs mainly). Also, the authors should make use of tables to better describe their findings and make information more reachable and meaningful for the readers (for example, the list from lines 274 to 289).

Author Response

Reviewer 2:

Comments and Suggestions for Authors

The authors review ontology-based regression testing studies using the PRISMA methodology seems sound. However, there are major issues that the authors need to address to improve the quality of the article.

Comment 1:

* Line 37: Abbreviations in abstract are in the form <long-name> (<short-name>). In this line is in the form '<long-name>' short-name. The authors are suggested to change it.

Response 1:

Thank you for your comment about the use of long-name and short-name in the abstract. In response to this comment, we have improved the paper and particularly at line 37 by using only long name. However, SLR and PRISMA on the first appearance are fully defined.

Comment 2:

* Line 101 and 102: "To develop a complete testing ontology, we require a new testing approaches and methods". This seems so disconnected from the paragraph about Zhu and Zhang's work. Is this sentence something you (the authors of this SLR) state or is it something Zhu and Zhang state? I think the sentence and maybe its position in the paragraph must be reconsidered.

Response 2:

We highly appreciate you comment. This sentence is about the research by Zhu and Zhang, and therefore, it has been improved for connection between sentences in a paragraph. The proposed approach was focused on the issue of testing of integrating web services. The authors of this study proposed ontologies in the research to support the wide range of software testing activities.

Comment 3:

* Line 109: "system a topology"?
Sentence from lines 158 to 160 is hard to understand.

Response 3:

Thank you for this comment. In order to address this comment, we have revisited the concerned studies to improve this part of the paper.

Comment 4:

Figure 3 has a bad quality and could be improved. Some lines are duplicated and some line angles (corners) are unnecessary.

Response 4:

Thank you for this comment. In response to this comment, we have improved the visibility of Figure 3 by removing the duplicate and unnecessary line angles.

Comment 5:

Roadmap or road map?

Response 5:

Thank you for your comment. Although, both words are correct, but there is more common use of a single word. Therefore, roadmap is used in this revised file.

Comment 6:

* "We can see that the number of research studies has been limited to TCP in regression testing." However, Figure 4 only shows an aggregation of ontology-based RT approaches. Therefore, I am not sure that the reader can "see" that studies have been limited to TCP.

Response 6:

Thank you for your comment. In response to this comment, we have improved readability of the concerned text. Actually, we meant it that studies on TCP topics were in limited numbers in other than 2011-2012 years. We have revised the text below Figure 4 on page #10 in the revised file.

Comment: 7

* It seems that some paragraphs are so disconnected from the previous ones and that the flow of the narrative is a little bit chaotic.

Reply:

Thank you for pointing out the disconnected paragraphs. We have reread the manuscript thoroughly and improved its flow of contents.  

Comment: 7

* This reviewer thinks that the authors have done a great work in reviewing the literature and applying the PRISMA methodology. However, the contribution of their work is hidden due to poor connection between the units of thought (sentences not properly fit to paragraphs and connection between paragraphs mainly). Also, the authors should make use of tables to better describe their findings and make information more reachable and meaningful for the readers (for example, the list from lines 274 to 289).

Reply:

Thank you for your kind feedback. In response to this comment, we have improved the flow of contents and connection between paragraphs. This can be seen in the revised file. Moreover, we created three new tables for RQs (Table 2), QAC (Table 3) and summary of the ontology-based approaches (Table 4) in this systematic literature review. Table 4 is showing us the summary of proposed approaches regarding regression testing and use of ontologies.

Reviewer 3 Report

The paper in this version is very good  I suggest to a minor  change in the title to be as follows:

Ontology-based Regression Testing: A Systematic Literature   Review

Author Response

Thank you for your suggestions to improve the title of the paper. In response to it, we have improved the title of this paper as per your kind suggestions.

Round 2

Reviewer 2 Report

The authors have improved the quality of the manuscript. However, I feel that there are some phrases and words that do not convey the expected meaning. English language, style and most importantly, the argumentation of some claims, should be revised.

For example:

Lines 289-290: "It is credible if test case grouping is 289
supported by any proposed and evaluated criteria" .

  • "It" -> What are you referring to? Avoid pronouns, they hide what they substitute, and this reduces the meaning of the sentence. The meaning is harder to grasp.
  • "credible" -> I do not think this wording is appropriate.  Might it be that the sentence could be rephrased as: "Grouping high and low priority test cases is justified and plausible only if it is supported by previously evaluated criteria".

Line 299:

"Risk has not been attributed to the quality attributes of web systems" -> Any web system (this claim seems so hard, maybe it needs to be softened)? Or the web systems that this SLR has considered? (those that are associated with the reviewed papers)?

These two are some examples of poor style/argumentation. Please, review those and other claims and enhance their style and argumentation.

Author Response

Thank you for your comments and suggestions about the English language and style. In response to your valuable comments, we have corrected the mentioned phrases, words, and argumentation of claims. In addition, we have thoroughly read the paper and corrected the language of manuscript wherever we found it necessary. We have displayed these changes or modifications in the manuscript via track changes.

Round 3

Reviewer 2 Report

English language and style has been improved. Arguments and claims are also clearer.

Still, minor grammar/spell checks are required.

Line 145:

Although the adaptive approach of risk estimation and test case ranking is of great importance, it was better if researchers discussed static web services regarding ontology-based TCP. -> "it was" -> "it would be"?

Other minor spell checks such as:

* rsisks -> risks

* arisk -> a risk

among others...

Author Response

Thank you for your comments and suggestions about the minor grammatical/spell check. In response to your valuable comments, we have corrected the minor grammatical errors. We have displayed these changes or modifications in the manuscript via track changes. Between this we have another round of grammar/spell checking of our submission and corrected wherever we found any error.